# A Systematic Review of Isotopically Measured Iron Absorption in Infants and Children Under 2 Years

**DOI:** 10.3390/nu16223834

**Published:** 2024-11-08

**Authors:** Samantha Gallahan, Stephanie Brower, Hannah Wapshott-Stehli, Joelle Santos, Thao T. B. Ho

**Affiliations:** 1Morsani College of Medicine, University of South Florida, Tampa, FL 33602, USA; samanthag20@usf.edu (S.G.); brower23@usf.edu (S.B.); 2Department of Pediatrics, Morsani College of Medicine, University of South Florida, Tampa, FL 33602, USA; wapshott@usf.edu; 3College of Arts and Sciences, University of South Florida, Tampa, FL 33602, USA; joellesantos@usf.edu

**Keywords:** iron, iron absorption, iron isotope, children, infant, preterm infant, nutrition, iron supplements

## Abstract

Background: Iron is an essential element for critical biological functions, with iron deficiency negatively affecting growth and brain development and iron excess associated with adverse effects. The goal of this review is to provide a comprehensive assessment of up-to-date evidence on iron absorption measured isotopically in children, preterm infants, and full-term infants, up to 24 months of age. Methods: Search databases included Pubmed, Cochrane, Web of Science, and Scopus from a date range of 1 January 1953 to 22 July 2024. The included articles were experimental studies with iron absorption outcomes measured by isotopic techniques. The risk of bias was assessed using the Cochrane Risk of Bias Tool. Results: A total of 1594 records were identified from databases, and 37 studies were included in the quality review with a total of 1531 participants. Article results were grouped by study commonality: absorption and red blood cell incorporation, type of milk feedings, additives to improve absorption, how and when to supplement with iron, and iron forms and complimentary foods. Conclusions: The results from this review support the current recommendations of oral iron supplementation. Iron from breast milk has high bioavailability, and unmodified cow’s milk reduces iron absorption. Supplemental iron is required at 4–6 months for healthy, full-term infants and sooner for preterm infants. Ascorbic acid increases iron absorption in full-term infants and children. Lactoferrin and prebiotics are promising candidates for enhancing iron absorption, but they require further investigation. Research evidence of iron absorption mechanisms and modulating factors in preterm infants is limited and should be a research priority.

## 1. Introduction

Iron is an essential element in human biology for oxygen transport, the immune system, central nervous system development, and DNA synthesis [1]. Iron deficiency (ID) is the most common nutritional deficiency worldwide, with the World Health Organization (WHO) estimating that 42% of children globally are iron deficient. Iron is in especially high demand during infancy and toddlerhood to support rapid neurodevelopment and physical growth, making this population particularly vulnerable [1]. ID in the central nervous system affects neurotransmitter development, myelination, and memory function [2,3]. Multiple studies show an association between iron deficiency anemia (IDA) in infancy and later cognitive deficits, potentially up to a 5–10 point deficit in intelligence quotient [3,4]. Preterm infants, born <37 weeks, are especially vulnerable to ID because they experience high iron demand with low iron acquisition. They are born with lower iron stores than full-term infants due to an incomplete maternal–fetal iron transfer occurring during the third trimester (after 27 weeks). In addition, preterm infants have high iron demand due to rapid postnatal growth and high iron loss from phlebotomy after birth [4,5].

Most of the molecular mechanisms of iron absorption during infancy and early childhood have been studied in animal models, with little confirmation in humans [6]. Iron is absorbed in the duodenum and proximal jejunum in adults, and there is evidence of iron absorption throughout the intestine of neonatal rats [7,8]. The most common form of non-heme iron from diet is ferric iron, which can be absorbed by receptor-mediated endocytosis or converted to ferrous iron, the iron form absorbed by the enterocytes. Ferric iron is converted to ferrous iron in the intestinal lumen due to environmental pH or using duodenal brush border membrane ferrireductases and other reductases [9,10]. Ferrous iron is taken up by mature enterocytes via the divalent metal iron transporter 1 (DMT-1) at the apical membrane [11]. Heme iron, found in animal products, is more readily absorbed than non-heme iron through receptor-mediated endocytosis [12]. Intracellular iron can be stored bound to ferritin protein or transported across the basolateral membrane into circulation via ferroportin-1 export protein (SLC40A1) [7]. In adults, the liver-produced hormone hepcidin inhibits this export by binding to ferroportin to cause its internalization and subsequent degradation [13]. The regulatory mechanism of hepcidin in preterm and full-term infants is less certain, but recent studies seem to show it is intact [14,15,16].

Considering the importance of iron homeostasis in young children and the consequences of ID on neural and behavioral development, iron absorption and factors that influence iron absorption are important areas of research. The goal of this systematic review is to provide a comprehensive assessment of current evidence on iron absorption measured isotopically in children, preterm and full-term infants, up to 24 months of age.

## 2. Materials and Methods

We conducted a systematic literature review according to the Preferred Reporting Items for Systematic Reviews and Meta-Analyses guidelines (PRISMA) [17]. The project was registered in the Center for Open Science at https://osf.io/5ayfp, accessed on 6 November 2024; however, the review protocol was not included on this registry. All prospective study designs were considered. We excluded review articles, case reports, articles presenting only secondary analyses, conference proceedings, and protocols. The study’s primary outcomes needed to include iron absorption measured by serum isotopic methods. Other outcomes include iron status measured by serum ferritin and/or other hematological indices. We included both randomized and non-randomized studies because some studied population conditions and interventions that cannot be randomized, such as the choice between breastfeeding and formula or when to wean off breastfeeding. There was also no restriction on the timeframe to capture the most complete picture of the research evidence available on this topic. Only articles available in English were selected for practical purposes. To be included, it was also required that articles were published with results.

The literature search strategy was developed by the first authors along with a professional medical research librarian. The search was intentionally broad to minimize the risk of overlooking potentially relevant studies. Multiple information sources were searched. These included Pubmed, Embase, Cochrane, Web of Science, and Scopus. The search date range was from 1 January 1953 to 22 July 2024 (70 years of data). Search terms included controlled vocabulary when appropriate and included terms related to children, iron absorption, and expected outcomes, including hepcidin, prohepcidin, liver-expressed antimicrobial peptide, isotope label, zinc protoporphyrin, hematocrit, hemoglobin, and ferritin. The complete list of search terms is available in the Appendix A. We also manually scanned the citations of included studies for relevant articles and references from similar systematic reviews in case they were missed during indexing. The software Rayyan (https://www.rayyan.ai, accessed on 6 November 2024) was used to manage records and to assist in study selection logistics and planning. After study selection was finalized, Microsoft Excel documents as well as Endnote were used for management of studies and data.

Each title and abstract were screened using the eligibility criteria by the 2 first authors independently, with the senior author to review for any conflicting responses. This method was also used for final inclusion screening of the full-text articles. Two first authors (S. B. and S. G.) reviewed all the included articles and then grouped them into categories (sometimes multiple) based on the broad goal of each study. After this, authors reviewed each study and extracted the following components: title, author(s), country, publication year, sample size, age, patient selection criteria, study design, length of study, intervention(s), and outcome(s).

The Cochrane Risk of Bias Tool was used to assess the bias of individual studies [18]. This process was performed at the study level by two first authors independently. The bias assessment was taken into consideration when presenting the studies and drawing conclusions. For result presentation, we grouped articles by their primary and secondary outcomes in tables, then by intervention types and studied populations. A study can be assigned to more than one group or table.

We did not conduct a meta-analysis due to the heterogeneity in populations and in outcomes. Qualitative analysis was done using GRADE (Grading of Recommendations, Assessment, Development, and Evaluations) guidelines [19]. The authors deemed meta-analysis and GRADE performances would not enhance the certainty of evidence with the included studies. However, the quality of evidence and risk of bias were considered in conclusion statements.

## 3. Results

### 3.1. Study Selection

A total of 1594 articles were identified through the database search. After removing 281 duplicates, 1313 articles were screened by title and abstract. Through this screening step, 1192 articles were excluded by studied population (n = 411), study design (n = 359), intervention (n = 152), outcomes (n = 100), and publication type (n = 170). A total of 121 articles underwent a full-text review. This full-text review process removed 84 articles for the studied population (n = 13), study design (n = 4), intervention (n = 37), outcomes (n =3), publication type (n = 18), and for not being in English language (n = 9) (Figure 1). A total of 37 articles were included in the final qualitative review.

### 3.2. Study Characteristics

The characteristics of the 37 articles included in the final qualitative analysis are presented in Table 1. These articles present the body of research over a span of over 60 years (1963–2023) and from 17 countries. There is a wide range of sample sizes, from 6 to 364 subjects with a median of 30 subjects. From study design classification, all included studies were clinical trials as defined by the National Institutes of Health. There were six (16%) studies that included data from preterm infants.

### 3.3. Quality of the Evidence

The most common sources of bias, summarized in Appendix A, were from the randomization process, missing outcome data, and measurement of the outcome. The risk of bias tool for clinical trials showed 60% (22/37) studies without randomization and the majority of studies with some concerns for missing outcome data due to high dropout rates or some risk of measurement of the outcome due to small sample sizes.

### 3.4. The Measurement of Iron Absorption and Incorporation

Iron intake and status can be estimated by using changes in the total body iron calculated by hemoglobin iron and body storage iron. Body storage iron can be measured directly with bone marrow aspirates and liver biopsies or estimated with serum ferritin, the storage form of iron [57]. Stable isotope techniques are necessary to measure iron absorption in humans. Therefore, we only included studies using isotopic techniques in this review. Iron absorption has been measured in a diverse number of term and preterm infants.

#### 3.4.1. Full-Term Infants

When ^57^Fe and ^58^Fe labeled isotopes were added to infant formula, the absorption of the two isotopes was not significantly different in 13–25-week-old infants [38]. Using ^57^Fe labeling, a study showed anemic Gambian infants between 14 and 20 months had a 3.8-fold increase in iron absorption but a 3.4-fold increase of iron loss while on daily 12 mg iron supplementation compared to when off iron supplementation. This means about 72% of absorbed iron was lost during the oral iron supplementation period [47]. It has also been observed that isotope excretion in feces occurred predominantly during the first 4 days and continued beyond 7 days after dosing [32]. When 9 infants at 3–10 weeks and 9 infants at 5–7 months were studied, mean retentions of ^58^Fe isotope were 31.2% and 26.9% in the younger infants versus 35% and 32.5% in the older infants at 4 and 11 days, respectively. The erythrocyte incorporation at 14 days was 5.2% of the dose in the younger infants and 12.5% in the older infants [32]. Measuring ^58^Fe directly in circulation, Fomon et al. reported a suitable method for within-subject comparison of iron availability from dietary intake [27]. Using isotopic techniques, it was found that in well-nourished, predominantly formula-fed, iron-sufficient infants, iron stores measured by serum ferritin were inversely correlated with the percentage of iron dose entering circulation [50].

#### 3.4.2. Preterm Infants

In preterm infants, the hemoglobin incorporation rate correlated significantly with the rate of growth and erythropoiesis, and they absorb and utilize iron at higher rates during the first 10 weeks after birth than [35]. Preterm infants born at 24–33 weeks at 4 weeks postnatal age had an iron absorption rate of 41.6% and an erythrocyte incorporation rate of 12% at 15 days after the first ^58^Fe dose [26]. McDonald et. al. studied iron absorption and red blood cell incorporation of iron from premature infant formula compared to stand-alone supplements between feedings. They found that red blood cell incorporation of iron from supplementation was slightly better than iron from formula [40]. Erythrocyte iron incorporation can also differ between enteral and parenteral iron administration in preterm infants. Zlotkin et al. found that only 17.8% of the intravenous infused iron dose was incorporated into hemoglobin, while 26% of the enteral iron dose was incorporated into red blood cells by day 15 in very low birth weight preterm infants [56]. Appendix A summarizes these studies.

### 3.5. Iron Absorption Based on Milk Feeding Types

#### 3.5.1. Breast Milk- and Cow’s Milk-Based Feedings

Iron in breast milk is more bioavailable than in cow’s milk-based formula or homemade cow’s milk feeding. This means a greater percentage of breast milk iron is absorbed than iron from other milk feeding types. At 6 months of age, full-term infants fed breast milk attained greater iron stores than did those fed a home-made cow’s milk formula due to a higher iron absorption rate from breast milk (49%) than from home-made cow’s milk formula (10%) [45]. Fomon et al. showed that 2-month-old full-term infants breastfed without iron supplementation had higher erythrocyte incorporation (20%) than infants fed low-iron formula (1.8 mg/L iron) (6.9%) [29]. This highlights the higher bioavailability of iron from breast milk compared to cow’s milk-based feeds. In 2-month-old full-term infants given a daily vitamin–iron supplement (7.5 mg iron daily), Fomon et al. found that mean erythrocyte incorporation was higher (7.8% of the dose) for breast-fed infants (n = 14) than (4.4% of the dose) for formula-fed infants (n = 15) [30]. Cow’s milk has low iron content (0.5 mg/L) and can also reduce iron absorption. Cow’s milk can inhibit the absorption of inorganic iron but not hemoglobin iron [36].

#### 3.5.2. Iron Given With or Without Feeds

The timing of the iron supplement relative to breast milk feeding can influence the absorption. Iron absorption of an ^58^Fe isotope supplement given with human milk was not different at 5–6 months vs. 9–10 months in full-term infants [37]. Those given iron without milk absorbed 19.2% at 5–6 months and 25.8% at 9–10 months. Those given iron with breast milk absorbed 42.6% at 5–6 months and 51.9% at 9–10 months. However, this study was not designed to directly compare the absorption between iron supplements given with and without the presence of breast milk [37].

#### 3.5.3. Formulas with Different Amounts of Fortification

The amount of iron in infant formulas varies. Low intake of iron may not prevent ID, and high intake of iron may negatively affect copper and zinc absorptions. Healthy infants 11–13 months require at least 7 mg/L of iron in formula to prevent ID [46]. Fomon et al. showed that red blood cell iron incorporation was not different in infants fed formulas fortified with 12 mg/L vs. 8 mg/L of iron [31]. On the other hand, the amount of fat, carbohydrates, or acidification of the formula did not influence iron absorption [48]. Table 2 summarizes these studies.

### 3.6. Additives to Improve Iron Absorption

Several additives can improve oral iron absorption. Higher iron absorption with fortification of ascorbic acid (100–800 mg/L) in formulas was observed in infants aged 5–18 months [48]. Similarly, iron absorption was greater when an iron supplement was given with juice containing ascorbic acid than with cow’s milk in 1-year-old children after weaning off formula [20]. This study also suggests that there is a dose-dependent effect of ascorbic acid, where a 3:1 or 4:1 ratio of ascorbic acid–iron may be optimal for iron absorption [20]. Lactoferrin is another additive that has been tested for optimizing iron absorption. The addition of apo-lactoferrin (iron-free form) but not holo-lactoferrin (iron-loaded form) to a test meal containing ferrous sulfate (FeSO_4_) significantly increased iron absorption by 56% [41]. Galacto-oligosaccharides (GOS) for 3 weeks, a prebiotic, increased iron absorption by 62% in 6–14 month-old-infants fed with micronutrient powder containing ferrous fumarate and sodium iron ethylenediaminetetraacetic acid (NaFeEDTA) [43]. GOS did not improve iron absorption of FeSO4 form [43]. On the other hand, adding a prebiotic GOS to a single test meal did not affect iron absorption from micronutrient powder containing FeSO4 or ferrous fumarate and NaFe EDTA [42]. Alpha-lactalbumin and casein-glycomacropeptide as additives in formula did not change iron absorption in infants from 4–8 weeks to 6 months of age [49]. Table 3 summarizes these studies.

### 3.7. Erythropoiesis Stimulating Agent and Iron Absorption

Erythropoiesis-stimulating agents, such as recombinant human erythropoietin (r-HuEPO), have been studied for their impact on iron absorption, especially in the preterm infant population. In a small study (n = 14) of preterm infants born <31 weeks and <1250 g, r-HuEPO (500 U/kg/wk) given during the first week after birth increased iron erythrocyte incorporation compared to placebo (4.4% vs. 2.0%, *p* = 0.013), but this was not observed with r-HuEPO given at 4 weeks. This study showed no effect of r-HuEPO on enteral iron absorption [54]. Similarly, higher doses r-HuEPO (2100 U/kg/wk) increased erythropoiesis but not enteral iron absorption, and the high dose increased iron incorporation into red blood cells only when given with oral and intravenous iron and not with oral iron alone [55] (Table 3).

### 3.8. Iron Supplement Doses, Timing, and Frequency

Iron supplement dose and frequency have been studied to optimize iron absorption, increase compliance, and prevent ID. In iron-sufficient 6-month-old infants, iron absorption was similar among infants fed high iron formula, low iron formula, or given iron drops with no iron formula. However, after 45 days on supplements, the high iron formula group had higher hemoglobin than the iron drops group, and serum ferritin levels were higher in the high iron formula and iron drops groups than the low iron formula group [50]. Different doses of iron sprinkles, 30 mg vs. 45 mg, added to maize-based food did not affect iron absorption [51].

Intestinal iron absorption can vary by postnatal age. Iron absorption was low and similar among healthy breast-fed, iron-supplemented, and unsupplemented infants at 6 months [25]. However, at 9 months, iron absorption from human milk was significantly higher in un-supplemented compared to iron-supplemented infants (36.7% vs. 16.9%). Iron absorption at 9 months was correlated with iron intake but not with iron status [25]. In 12–23-month-old Malawian toddlers with malarial infection, oral FeSO_4_ supplements immediate after malaria treatment or delay for 2 weeks resulted in similar iron absorption and hemoglobin levels [34].

Iron in the form of micronutrient powder given daily to 5–14-month-old infants in Kenya showed no significant difference in iron absorption nor plasma hepcidin when given in the morning compared to the evenings [52]. In the same study, daily dosing increased serum hepcidin levels and decreased iron absorption compared to every-other-day dosing. Every-other-day and every-third-day dosing did not increase serum hepcidin or decrease iron absorption [52]. Table 4 summarizes these studies.

### 3.9. Iron Forms and Methods of Delivery

Several studies have compared the effects of iron forms and delivery methods on iron absorption and status. Commonly, iron fortification is given as FeSO_4_, ferrous fumarate, ferrous ascorbate, or NaFeEDTA. Other forms of iron have been studied, including sodium iron pyrophosphate, ferric orthophosphate, iron glycine, ferric polymaltose, ferric ammonium citrate, and bovine hemoglobin. Iron can be delivered through fortification of formula and baby foods or independently as a liquid or powder supplement.

#### 3.9.1. Different Forms of Iron

Chavasit et al. exposed 8–24-month healthy children to ferric ammonium citrate or a combination of FeSO_4_ and NaFeEDTA at 2:1 ratio in quick-cooking rice and compared to a reference FeSO_4_ group. The results showed the mean fractional iron absorption was high in both groups: 5.8% in the ferric ammonium citrate group and 10.3% in the NaFeEDTA group. The relative bioavailability of iron compared to the reference FeSO_4_ group was 83% for ferric ammonium citrate and 145% for NaFeEDTA group [23]. Another study showed fractional iron absorption from FeSO_4_ was 40% higher than from ferrous fumarate plus NaFeEDTA [42]. Bioavailability of iron in NaFeEDTA and FeSO_4_ with ascorbic acid was similar when being fortified in wheat and soy-based complementary baby foods [24]. Sodium iron pyrophosphate and ferric orthophosphate were poorly absorbed from infant cereal, cow’s milk, and soy-based formulas, while FeSO_4_ in infant cereal or formulas was adequately absorbed to meet dietary needs [44]. Fox et al. studied the absorption of glycine-chelated iron compared to FeSO_4_ added to complimentary baby foods with and without the presence of dietary inhibitor phytate in 9-month-old infants. There was no difference in absorption between iron glycine and FeSO_4_ in low- and high-phytate complimentary baby foods. Both had lower bioavailability in high-phytate baby foods such as whole-grain cereal [33]. Heme iron from foods is generally absorbed better than inorganic iron. However, iron absorption of heme-fortified rice cereal was 14.2% compared to ferrous ascorbate reference with an absorption rate of 38.5% [22]. Micro-encapsulated ferrous fumarate with a lipid coating had a lower absorption rate than non-encapsulated form when they were added to complimentary baby foods [39].

#### 3.9.2. Complimentary Foods

The impact of complimentary baby foods on iron absorption has been studied. In Jamaican infants from 5 months to 2 years, the mean absorption of iron from maize (4.3%) and soybean (9.4%) was lower than from ferrous ascorbate drops (28.5%) [21]. The erythrocyte iron incorporation of FeSO_4_ (0.85 mg of ^58^Fe)-fortified Mead Johnson Enriched Baby Food (0.05 mg) and vegetables and beef baby food (0.08 mg) were low. The mean iron erythrocyte incorporations were higher in FeSO_4_-fortified rice cereal with fruits (0.15 mg), grape juice (0.14 mg), and ferrous fumarate-fortified Mead Johnson Enriched Baby Food (0.18 mg) [28].

The influence of whole grains versus refined flour on iron absorption has been studied. In 7–12-month-old infants with hemoglobin >100 g/L, similar absorption of ferrous fumarate mixed in rice-based and wheat-based complementary baby foods [39]. In 6–14-month-old Malawian children with high prevalence of anemia and ID, fractional iron absorption from whole grain oat fortified with ferrous bisglycinate was significantly lower than the absorption of ferrous fumarate from the reference refined wheat meal (7.4% vs. 12.1%). Among the whole grain preparations, fractional absorptions of ferrous fumarate from wheat with lentil and wheat with chickpeas were significantly higher than from oat (15.8%, 12.8%, and 9.2%, respectively) [53]. Appendix A summarizes these studies.

### 3.10. The Influence of Baseline Status on Iron Absorption

The baseline iron status of children can affect iron absorption, as the downregulation of hepcidin in ID is thought to lead to increased iron absorption. Heinrich et al. found that the absorption was higher in infants with depleted iron stores compared to infants with normal iron stores for both inorganic iron ^59^Fe (26% vs. 18%) and hemoglobin iron (8.3% vs. 4.8%) [36]. Tondeur et al. found similar findings, with iron absorption higher (8.25%) in infants with IDA than in infants with ID without anemia (4.48%) or in iron-sufficient (4.65%) infants [51].

## 4. Discussion

This systematic review summarizes over 60 years of research on iron absorption measured by isotopic methods (Appendix A). Although the studies are heterogeneous in tested interventions, population characteristics, and measured outcomes, there are several consistent findings. Iron in breast milk has higher bioavailability than iron from formula and unmodified cow’s milk [30,45,58,59]. Unmodified cow’s milk reduces net iron absorption in infants [36,45]. Iron content greater than 4–12 mg/L in formula does not provide additional benefits in iron absorption and iron status but can potentially have negative effects on other metals’ absorption and metabolism [31,46]. Ascorbic acid improves iron absorption, while lactoferrin and GOS require further studies [20,41,42,43,48]. These findings have been further tested in non-isotopic clinical trials discussed below.

### 4.1. Current Guidelines

Clinical studies have shaped the current recommendations (Figure 2) on iron supplementation from various pediatric scientific societies, but more research evidence is needed. For the first 4–6 months after birth, with high bioavailability, 0.27 mg/day of iron from breast milk is appropriate for healthy, full-term infants with adequate iron stores. However, breast milk alone is not an adequate source of iron after 4–6 months after birth for full-term infants and sooner for preterm infants [60,61,62]. Evidence strongly supports that unmodified cow’s milk reduces iron absorption. Thus, all pediatric societies recommend delaying unmodified cow’s milk introduction until after 1 year of age in both term and preterm infants and avoid excessive daily intake after 1 year of age [4,36,46,57,58,59,60,63,64,65,66].

As mentioned above, human milk has a low amount of highly bioavailable iron. The mechanistic explanation for this high bioavailability is still unknown. Current research suggests the higher bioavailability is due to iron-containing bioactive proteins in breast milk, including lactoferrin, transferrin, xanthine oxidase, and ferritin. Lactoferrin has been extensively studied as a facilitator of iron absorption, and the results have been inconsistent [41,67,68]. Apo-lactoferrin, the iron-free form of lactoferrin, had greater effects on iron absorption than the iron-saturated form of lactoferrin (hololactoferrin) [41]. Formulas with and without fortification of bovine lactoferrin have been compared to breast milk. Breast-fed infants had significantly higher serum ferritin levels compared to infants fed unfortified formula at 30 and 90 days, and they had higher serum ferritin levels compared to infants fed lactoferrin-fortified formula only at 30 days [67]. Infants fed bovine lactoferrin-fortified formula starting at 4–6 months for 3 months had higher iron indices, such as serum ferritin and total iron binding capacity, and lower incidences of ID and IDA compared to those fed with unfortified formula [68]. On the other hand, using fecal isotope measurement, 7-day-old infants had similar iron retention given bovine lactoferrin vs. ferric chloride over 3 days [69]. The effects of lactoferrin on iron absorption may be indirect and time-dependent and require further investigations.

Iron fortification of 4–12 mg iron/L in formula is required to prevent ID and IDA, with higher doses yielding little increase in absorption. The studies covered in Table 2 agree with this metric. Iron supplementation higher than 12 mg/L in formula may also decrease the absorption of other trace metals and increase the risk of oxidative stress from Fenton and Haber-Weiss reactions, but studies on these do not currently agree [46,61,70,71]. For example, Friel et al. saw an increase in glutathione peroxidase levels and number of respiratory tract infections in infants receiving formula with 20.7 mg iron/L compared to 13.4 mg iron/L, while Power et al. only saw a decrease in serum zinc and no increase in infection when infants were given 8.3 mg Fe/100 g vs. 40 mg Fe/100 g [70,71]. Though it was worth mentioning infants in the Friel et al. study were low birth weight and infants in the Power et al. study were not. This highlights the importance of the infant baseline characteristics in iron absorption and metabolism.

For preterm infants, the evidence on iron absorption and requirement is lacking; therefore, societies’ recommendations are sparse and inconsistent [4,72]. Unlike full-term infants, preterm infants require iron supplementation before 4–6 months to avoid a negative iron balance [62]. This is due to interrupted accretion in the womb, high demand for rapid growth, and high iron loss after birth. However, the optimal iron dose and timing are unknown, and these likely depend on the birth gestational age, baseline iron stores, postnatal age, and other biological conditions. Due to their vulnerability, isotopic studies in preterm infants are older and with smaller sample sizes. The absorption rate of oral iron measured by isotope studies ranges 19–42%, and the red blood cell incorporation rate ranges 4–15% [26,35,56,62]. The iron absorption rate in preterm infants is generally driven by growth and body demand but not consistently by dose [35,62]. The timing of enteral iron supplementation in preterm infants born <1500 g was studied in Sankar et al. Starting iron at 14 days or at 60 days yielded no statistical significance in iron stores and hematocrit at 2 months, number of blood transfusions, and common neonatal morbidities [73]. Among low birth weight (1500–2000 g) infants given iron drops at 4–6 months, iron status was higher in an exclusively breast-fed group compared to those given complementary iron-fortified baby foods while continuing breast feeding [74]. As preterm infants are most vulnerable to ID, oxidative stress, and abnormal gut microbiota development, future studies should investigate approaches to maximize iron absorption and minimize adverse effects from excess iron.

The use of exogenous erythropoietin is another practice in the care of preterm infants that varies. Exogenous erythropoietin increases iron incorporation into red blood cells; it does not improve intestinal iron absorption [55,75,76]. Thus, while r-huEPO may prevent anemia, oral iron supplementation in conjunction with EPO may not be able to maintain adequate body iron stores and iron supply to the brain. In non-transfused preterm infants on r-huEPO, intravenous iron was more effective in improving iron status than oral iron; however, higher plasma malondialdehyde (a measure of oxidative stress) was observed immediately after iron infusions [76]. Preterm infants receiving high-dose r-huEPO therapy at 1200 IU/kg/week had decreasing ferritin levels at both 6 mg/kg/day and 12 mg/kg/day of iron supplements [75], which is consistent with the earlier isotopic studies that show no increase in intestinal iron absorption.

An infant’s baseline iron status in general can also affect iron absorption and should be considered in preventive care [36,51]. Iost et al. determined the repletion rate of hemoglobin in infants presenting with different degrees of IDA using ferrous iron amino acid chelate-fortified cow’s milk. The highest rates of repletion were in the severely anemic group, whereas children with initially normal hemoglobin concentration showed no change [77]. Reeves et al. randomized non-anemic, healthy, full-term infants (Hct >10.5 gm/dL) to 30 mg of iron daily or placebo and found that the improvement in Hb was low and the same between the groups without history of prior infections. However, the repletion rate was greater in infants that had been seen recently for infections [78]. Infants with protein malnutrition can have significant differences in serum iron after iron administration, but there were no differences in hematocrit or transferrin saturation [79].

### 4.2. Iron Absorption Optimization

To optimize iron absorption, several studies have investigated forms of iron and the effects of other compounds in the diet. Even though other iron forms, including ferric fumarate, iron glycine, ferric pyrophosphate, and NaFeEDTA, have been absorbed sufficiently in some studies, FeSO4 remains the most appropriate and commonly used form of iron supplement in children as iron drops or fortification in formula and baby foods [23,24,33,80]. Ascorbic acid is well known to improve iron bioavailability [20,48]. The presence of ascorbic acid increases the absorption of non-heme iron by allowing it to remain ferrous and soluble in the duodenum. Polyphenols and phytates, however, reduce intestinal iron absorption [33]. By complexing with iron, they abolish iron bioavailability because the body is unable to digest these components [81]. In infants and toddlers at risk for ID with history of low iron diet, 8 weeks of 18 mg/day of vitamin E did not improve iron absorption [82]. When exogenous supplements are encapsulated, emulsified, chelated, or nanoparticulated, the bioavailability of iron can sometimes be improved [81]. However, results from studies involving different complimentary baby foods are difficult to generalize [21,28,53].

Prebiotics like GOS are commonly added to formulas for potential contribution to healthy gut microbiota [83]. GOS has been studied as a potential enhancer of iron absorption. GOS increased absorption of iron fumarate and NaFeEDTA but not FeSO_4_ after 3 weeks of supplement. However, a one-time dose of GOS did not show a benefit [42,43]. A recent clinical trial showed that prebiotics in iron-fortified cereal increased iron absorption and improved gut microbiota after 3 weeks [84]. A plausible explanation is that GOS promotes the growth of gut-beneficial lactobacilli and bifidobacterium, which can affect the gut environment, such as reducing pH and iron-sequestrating bacteria [83,85,86]. However, in comparing studies on the use of prebiotics in infants, it is important to consider that changing the gut microbiota is a gradual process, and the benefits on iron absorption may take time [87]. Therefore, more studies are needed to clarify the benefits, optimal dosage, and duration of GOS supplementation, as well as the form of iron being co-administered.

Daily iron dosing is the most common recommendation. However, due to poor compliance and gastrointestinal adverse effects, less frequent dosing schedules have been studied. Intermittent oral iron supplements reduce hepcidin production to favor higher iron absorption in short-term experimental studies of children and women, but long-term effects of intermittent oral iron supplements on iron status and anemia prevention in term and preterm infants are not available [88]. In 5–14-month-old infants with high prevalence of anemia, baseline iron absorption decreased and plasma hepcidin increased with daily dosing, and these changes were not observed with every-other or every-third-day dosing [52]. Once a week dosing was effective in preventing IDA compared to no iron supplementation and not worse than daily dosing in 1–2-year-old children [89]. A Cochrane review of 33 clinical trials up to May 2011 shows that daily iron is more effective than intermittent iron in preventing anemia in children less than 12 years old [90]. More clinical trials are needed to investigate the long-term effectiveness of intermittent iron dosing in preventing ID in both term and preterm infants.

Our primary goal for this systematic review was to evaluate research evidence on iron absorption in infants and children up to 2 years of age. The search terms were broad, and the inclusion criteria for included articles focused on the isotopic measurements of iron absorption and iron status as outcomes. This approach allowed us to broaden the search to minimize missing qualified articles while maintaining the focus on the topic of iron absorption. The review was not restricted to randomized controlled trials; however, all included articles are clinical trials. The heterogenicity of the studies’ inclusion criteria and outcomes did not allow a meta-analysis that would increase the certainty of the conclusions. Additionally, we did not contact the authors of the included papers for further information, which could have provided greater insight. Including only articles written in English and published with full text could have left out a few impactful studies. Finally, this systematic review presents a historical shift in methodology and focus of clinical studies from what absorbed best to what and how to improve absorption in children, especially in preterm infants.

## 5. Conclusions

Results from this systematic review of research evidence on oral iron absorption in term and preterm infants and children up to 2 years of age support the current recommendations, but more research evidence is needed for preterm infants. Supplemental iron is required at around 4–6 months for healthy, full-term infants and sooner for preterm infants. Unmodified cow’s milk increases the incidence of ID and IDA, so it is not an appropriate nutritional source for infants less than 1 year. Ascorbic acid increases iron absorption in full-term infants and children. Lactoferrin and prebiotics are promising candidates for enhancing iron absorption, but further studies are required. Research evidence of iron absorption mechanisms and factors that influence iron status in preterm infants is limited. Since preterm infants have the greatest benefit from iron sufficiency and are most vulnerable to ID and excess iron in the gut, studies to increase iron absorption efficacy and reduce adverse effects from unabsorbed iron in this population should be a research priority.

## Figures and Tables

**Figure 1 nutrients-16-03834-f001:**
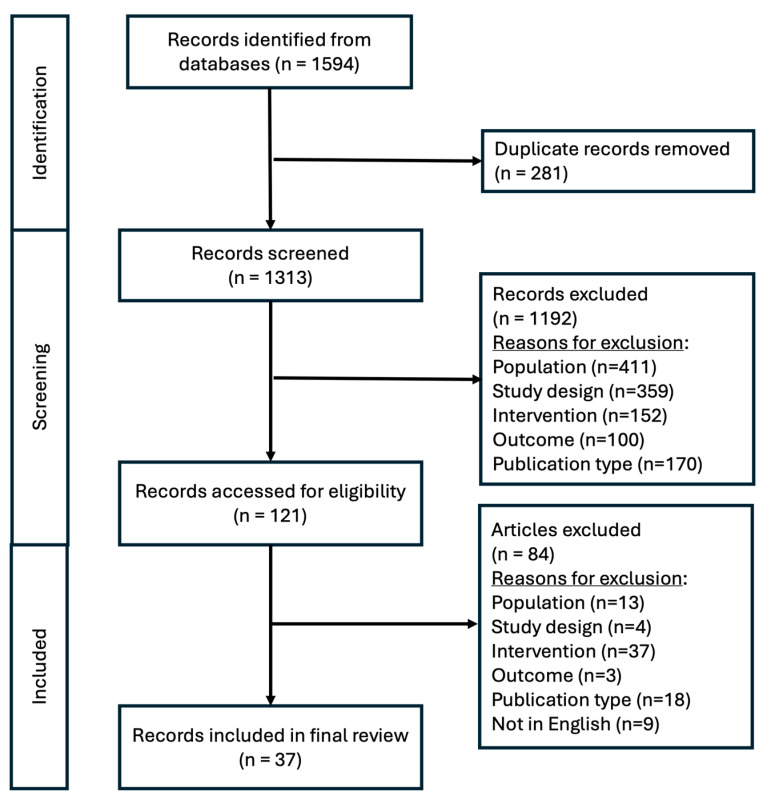
PRISMA Flow Diagram.

**Figure 2 nutrients-16-03834-f002:**
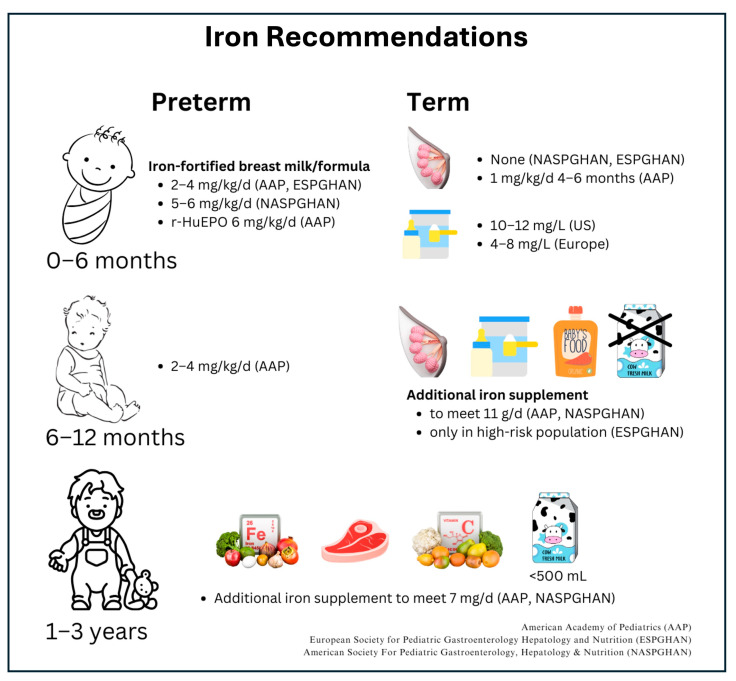
The current recommendations of iron intake for infants and children up to 3 years of age.

**Table 1 nutrients-16-03834-t001:** All I included articles by authors.

Authors	Year	Country	Sample Size	Population	Comparison and Testing
Abrams, et al. [20]	1996	USA	10	12–15 months	^57^FeSO_4_ with cow milk vs. ^58^FeSO_4_ in apple juice
Ashworth, et al. [21]	1972	Jamaica	42	5–24 months	Maize, boiled soya beans, and baked soya beans vs. ferrous ascorbate
Calvo, et al. [22]	1989	Chile	10	8–10 months	Hemoglobin-fortified cereal vs. ferrous ascorbate
Chavasit, et al. [23]	2016	Thailand	30	8–24 months	Rice fortified with ^58^Fe ammonium citrate vs. ^58^FeSO_4_ + NaFeEDTA
Davidsson, et al. [24]	2005	USA	11	18–27 weeks	NaFeEDTA vs. ^58^FeSO_4_ in wheat or soy-based foods
Domellöf, et al. [25]	2002	Sweden	25	4–9 months	Starting iron supplements at 4, 6 vs. 9 months
Ehrenkranz, et al. [26]	1992	USA	11	Preterm infants born <33 weeks	^58^FeSO_4_ plus ascorbic acid
Fomon, et al. [27]	1988	USA	9	112 days	^58^FeSO_4_ absorption
Fomon, et al. [28]	1989	USA	49	112 days	^58^FeSO_4_ in different foods
Fomon, et al. [29]	1993	USA	34	2–4 months	^58^FeSO_4_ in breastfed or low-iron formula-fed infants
Fomon, et al. [30]	1995	USA	30	56 days	^58^FeSO_4_ in breastfed and formula-fed infants
Fomon, et al. [31]	1997	USA	52	112 days	Formula with 8 mg/L vs. 12 mg/L iron
Fomon, et al. [32]	2000	USA	24	20–215 days	^58^FeSO_4_ for 11 days vs. 14 days
Fox, et al. [33]	1998	England	46	9 months	Iron glycine vs. FeSO_4_ in infant foods
Glinz, et al. [34]	2014	Malawi	48	12–24 months	FeSO_4_ immediately vs. 2 weeks after malarial treatment
Gorten, et al. [35]	1963	USA	14	Preterm infants	^59^FeCl_3_ at different doses
Heinrich, et al. [36]	1975	Germany	not stated	1–18 months	^59^FeSO_4_ and ^59^hemoglobin in diluted cow’s milk
Hicks, et al. [37]	2006	Peru	38	5–10 months	^58^Fe and ^57^Fe in breast milk or with juice
Kastenmaye, et al. [38]	1994	France	9	13–25 weeks	2.5 mg ^57^Fe and 0.6 mg ^58^Fe in formula
Liyanage and Zlotkin [39]	2002	Sri Lanka	39	7–12 months	Encapsulated vs. non-encapsulated ferrous fumarate in rice and wheat cereals
McDonald, et al. [40]	1998	USA	13	Preterm infants born 27–30 weeks and <1500 g	^54^Fe in formula, ^57^Fe in multivitamin drops, and ^58^Fe intravenous
Mikulic, et al. [41]	2020	Kenya	25	3–6 months	^58^FeSO_4_ apo-Lactoferrin vs. ^57^FeSO_4_ holo-Lactoferrin vs. ^54^FeSO_4_
Mikulic, et al. [42]	2021	Kenya	23	6–14 months	Iron with vs. without galacto-oligosaccharides
Paganini, et al. [43]	2017	Kenya	50	6–14 months	Micronutrient with and without galacto-oligosaccharides
Rios, et al. [44]	1975	USA	67	4–7 months	Different forms of iron in cereals and formula
Saarinen, et al. [45]	1977	Finland	45	6–7 months	^59^FeSO_4_ during breastfeeding, between breast feedings, and from home-prepared cow’s milk.
Saarinen and Siimes, et al. [46]	1977	Finland	30	11–13 months	Formula with 0.8 vs. 6.8 vs. 12.8 mg/L ^59^FeSO_4_
Speich, et al. [47]	2021	Gambia	22	14–20 months	12 mg/day ^57^FeSO_4_ followed by no iron
Stekel, et al. [48]	1986	USA	364	5–18 months	^59^FeSO_4_ in different formula preparations with and without ascorbic acid
Szymlek-Gay, et al. [49]	2012	Sweden	42	1–2 months	^58^Fe in formula with and without lactalbumin and casein-glycomacropeptide
Szymlek-Gay, et al. [50]	2016	Sweden	72	6 months	6.6 mg/day in formula vs. 1.3 mg/day in formula vs. 6.6 mg/day iron drops
Tondeur, et al. [51]	2004	Ghana	90	6–18 months	30 mg vs. 45 mg iron in maize-based foods
Uyoga, et al. [52]	2020	Kenya	78	5–14 months	Iron given at different dosing schedules
Uyoga, et al. [53]	2022	Malawi	30	6–14 months	Ferrous fumarate and ferrous bisglycinate in different baby foods
Widness, et al. [54]	1997	USA	14	Preterm infants born <31 weeks and <1250 g	^58^Fe with and without r-HuEPO
Widness, et al. [55]	2006	Austria	29	Preterm infants born <31 weeks and <1300 g	Oral polymaltose ^57^Fe vs. intravenous ^58^Fe with r-HuEPO
Zlotkin, et al. [56]	1995	Canada	6	Preterm infants born <1500 g	Intravenous ^57^FeSO_4_ and oral ^58^FeSO_4_

r-HuEPO: recombinant human erythropoietin.

**Table 2 nutrients-16-03834-t002:** Iron absorption from breast milk and cow’s milk-based feedings.

References	Population Characteristics	Interventions	Outcomes
Heinrich et. al. (1975) [36]	Full-term infants 1–18 mos old	^59^FeSO_4_ or ^59^hemoglobin w/diluted cow’s milk Measured absorption after 14 days	▪ Diluted cow milk reduced iron absorption from FeSO_4_ but not from hemoglobin iron in infants w/normal and depleted iron stores.
Saarinen et al. (1977) [45]	45 full-term infants 6–7 mos old No ID	Group 1: breast milk since birth–test ^59^FeSO_4_ during breastfeeding Group 2: breast milk since birth–test ^59^FeSO_4_ after 3-h fasting followed by 1-h fasting Group 3: breast milk from birth to 2 mos, then home-prepared cow’s milk formula. Test ^59^FeSO_4_ given fasting like group 2	▪ Groups 1 and 2 had higher serum ferritin at 6 mos than group 3; ▪ Breast milk promoted higher iron absorption than cow’s milk by 39% for group 1 compared w/group 3, despite group 3 having significantly lower iron stores.
Saarinen and Siimes (1977) [46]	30 full-term infants 11–13 mos old	Group 1: Formula fortified with 0.8 mg/L ^59^FeSO_4_ Group 2: Formula fortified with 6.8 mg/L ^59^FeSO_4_ Group 3: Formula fortified with 12.8 mg/L ^59^FeSO_4_	▪ Groups 2 and 3 had significantly higher absorbed iron amount than group 1; ▪ No differences between groups 2 and 3; ▪ No correlation was found between iron absorption and Hb levels, MCV, serum transferrin saturation, or serum ferritin.
Stekel et al. (1986) [48]	396 full-term infants 5–18 mos old	Different iron (10–19 mg/L FeSO_4_)-fortified formulas with and without ascorbic acid	▪ Mean absorption rate of ferrous ascorbate was 34.4% and FeSO_4_ from formula were 2.9–11.3%; ▪ Ascorbic acid ≥100 mg/L increased iron absorption by 2–3x; ▪ Plateau iron absorption with ascorbic acid >200 mg/L.
Fomon et al. (1993) [29]	34 full-term infants 2–4 months old	Group 1: Breast milk Group 2: Low iron formula (1.8 mg/L) At 56d old: 0.6–1 mg ^58^Fe given with ascorbic acid between feeds for 3 d. measurements were made at 70d and 112d.	▪ At 70d, erythrocyte incorporation was higher in group 1 than group 2; ▪ Erythrocyte incorporation was greater at 112d than at 70d in group 1 but not in group 2.
Fomon et al. (1995) [30]	30 full-term infants 8–16 wks old	Group 1: Breast milk Group 2: Low iron formula (1.8–1.9 mg/L) At 56 days: infants were given 0.5–1 mg ^58^FeSO_4_ for three consecutive days. 4 days after last dose, breastfed infants were given 7.5 mg/d FeSO_4_ and formula babies switched to a high iron formula (12 mg/L)	▪ Serum ferritin was lower in infants fed low iron formula, but hemoglobin was the same; ▪ A greater percentage of ^58^FeSO_4_ was incorporated in erythrocytes of breastfed infants than formula-fed infants when it was administered between feedings.
Fomon, et al. (1997) [31]	52 infants 112 days of age Were on formulas	Infants fed formulas with 8 mg/L vs. 12 mg/L FeSO_4_ ^58^Fe test meals for 3 days at 154 days	▪ No differences in iron incorporation into red blood cells.
Hicks et al. (2006) [37]	38 full-term infants born >2500 g 5–6 mos breastfed 9–12 mos breastfed and baby foods No history of iron supplement	D0: blood draws and breast milk collection D1: 150 μg ^58^Fe in breast milk given with 2-h fast before and after. D2: 2 mg ^57^FeSO_4_ given with 50 mg ascorbic acid in juice with 2-h fasting before and after D14: blood draws	▪ Age and baseline anemia status had no significant effects on iron absorption from breast milk and from juice; ▪ Serum ferritin was inversely correlated with iron absorption.

^57^Fe = stable isotope iron 57, ^58^Fe = stable isotope iron 58, ^59^Fe = radioisotope iron 59, mos = months, wks = weeks.

**Table 3 nutrients-16-03834-t003:** Studies of additives and erythropoietin to improve iron absorption.

References	Population Characteristics	Interventions	Outcomes
Stekel et al. (1986) [48]	364 full-term infants 5–18 mos old	Different ^59^FeSO_4_-fortified formulas w/and w/out ascorbic acid	▪ Ascorbic acid ≥100 mg/L increased iron absorption by 2–3x ▪ Plateau iron absorption with ascorbic acid >200 mg/L
Abrams et al. (1996) [20]	10 infants 12–15 mos old On formula for 6 mos then cow’s milk for 1–3 mos No history of anemia or iron supplement	^57^FeSO_4_ w/whole cow’s milk vs. ^58^FeSO_4_ w/apple juice	▪ Absorption of iron greater w/apple juice ▪ Iron absorption w/juice correlated w/serum ferritin
Paganini et al. (2017) [43]	50 infants 6–14 mos old	Iron (^57^Fe fumarate + Na^58^FeEDTA or ^54^FeSO_4_) w/and w/out GOS in maize porridge for 3 weeks	▪ GOS increased absorption of iron in form of Fe fumarate plus NaFeEDTA, but not FeSO_4_
Mikulic et.al. (2021) [42]	23 infants 6–14 mos old 87% iron deficient 70% anemic	^57^Fe fumarate + Na^58^FeEDTA or ^54^FeSO_4_ w/and w/out GOS All infants received all 4 test meals in maize porridge randomly on days 2, 3, 19, and 20.	▪ GOS had no significant effect on iron absorption ▪ Iron absorption from FeSO_4_ was higher than from Fe fumarate + NaFeEDTA w/and w/out GOS
Mikulic et al. (2020) [41]	25 infants 3–6 mos old	Group 1: ^54^FeSO_4_ Group 2: ^58^FeSO_4_ w/apo-lactoferrin Group 3: ^57^Fe holo-lactoferrin Iron forms mixed in maize porridge	▪ Iron absorption is higher in apo-lactoferrin group than holo-lactoferrin and ^54^FeSO_4_ groups ▪ No difference between holo-lactoferrin and ^54^FeSO_4_ groups
Szymlek-Gay et al. (2012) [49]	42 infants 1–2 mos old	Group 1: standard formula Group 2: α-lactalbumin enriched formula Group 3: α-lactalbumin enriched and casein-glycomacropeptide reduced formula All formulas had 4 mg iron/L From 4–8 wks to 6 mos	▪ α-lactalbumin and casein-glycomacropeptide did not affect iron absorption from formula
Widness et al. (1997) [54]	14 preterm infants Born <31 wks and <1250 g	With and without 6 wks of r-HuEPO (500 U/kg/wk), with 6 mg/kg/d FeSO_4_ ^58^Fe test dose given at d8 (early dosing) and d29 (late dosing)	▪ Erythrocyte incorporation was greater in EPO group after early dosing but similar to placebo in late dosing ▪ No change in erythrocyte incorporation over time in EPO group but increased significantly in placebo group ▪ No difference in iron absorption between the groups
Widness et al. (2006) [55]	29 preterm infants Born <31 wks and <1300 g at birth	Group 1: Oral ^57^Fe polymaltose complex Group 2: Daily oral ^57^Fe polymaltose complex and r-HuEPO (2100 U/kg/wk) Group 3: Daily oral ^57^Fe polymaltose complex, 2 mg/kg/d intravenous ^58^Fe sucrose and r-HuEPO (2100 U/kg/wk)	▪ EPO and intravenous Fe improved red blood cell Fe incorporation and erythropoiesis ▪ EPO improved erythropoiesis but not iron incorporation ▪ Iron absorption was the same in all groups

r-HuEPO = recombinant human erythropoietin, GOS = galacto-oligosaccharide, mos = months, wks = weeks.

**Table 4 nutrients-16-03834-t004:** Iron absorption and the timing, frequency, and doses of iron supplementation.

Categories	References	Population Characteristics	Interventions	Outcomes
Timing	Domellof et al. (2002) [25]	15 infants 4 mos old Exclusively breastfed up to 6 mos then partially breastfed till 9 mos	Group 1: 1 mg/kg/d iron from 4 to 9 mos Group 2: Placebo 4 to 6 mos and iron supplemented at 6 to 9 mos Group 3: Placebo 4 to 9 mos ^58^Fe in breast milk given at 6 and 9 mos to test absorption	▪ At 6 mos, iron absorption from human milk was low and similar among the groups ▪ At 9 mos, iron absorption from breast milk was higher in un-supplemented than supplemented infants ▪ There was a significant inverse correlation between dietary iron intake and absorption of iron from breast milk at 9 mos
	Glinz et al. (2014) [34]	48 infants with acute, uncomplicated malaria 12–24 mos old	30 mg FeSO_4_ daily for 8 wks immediately or 2 wks after malaria treatment	▪ Iron absorption on the first day of supplementation, 2 wks, or 8 wks in was not significantly different for both groups ▪ No significant difference in hemoglobin concentration between groups after supplementation period
	Uyoga et al. (2020) [52]	78 infants 66% with anemia 5–14 mos old	Study 1: Micronutrient powder w/12 mg FeSO_4_ given in maize porridge in the morning or afternoon	▪ Plasma hepcidin levels and iron absorption did not differ between the groups
Frequency	Uyoga et al. (2020) [52]	78 infants 66% with anemia 5–14 mos old	Study 2: Micronutrient powder w/12 mg FeSO_4_ given in maize porridge every day or every other day Study 3: Micronutrient powder with 12 mg FeSO_4_ given in maize porridge every other day vs. every third day	▪ Baseline iron absorption significantly decreased and plasma hepcidin increased w/daily dosed iron ▪ This effect was not seen when iron was dosed every other or every third day
Doses	Tondeur et al. (2004) [51]	90 infants 6–18 mos old	Microencapsulated ferrous fumarate iron sprinkles added to maize porridge on 3 consecutive days: 30 mg/d vs. 45 mg/d	▪ No significant effect of dose on iron absorption ▪ Absorption was significantly higher in infants w/IDA than infants with ID or sufficient iron ▪ No significant difference in absorption between infants w/ID and sufficient iron
	Szymlek-Gay et al. (2016) [50]	72 iron sufficient infants 6 mos old	FeSO_4_ 6.6 mg/d in formula, 1.3 mg /d in formula or 6.6 mg/d in drops for 45 days Test ^57^Fe and ^58^Fe given on study day 31 (at 7 mos of age).	▪ No differences in iron absorption and erythrocyte incorporation

mos = months, wks = weeks.

## Data Availability

The original contributions presented in this study are included in the article/Appendix A. Further inquiries can be directed to the corresponding author(s).

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
