# Peer review of "A Systematic Review of Isotopically Measured Iron Absorption in Infants and Children Under 2 Years"

_nutrients, 2024, doi:10.3390/nu16223834_

Round 1
Reviewer 1 Report
Comments and Suggestions for Authors
This review is thoughtful and of much interest to readership. The study logic is clear in its approach to complex topic of interest to societies who generate guidelines. The authors followed traditional PRISMA and other guidelines. There is clinical and scientific need to make the data both complete and understandable for the more casual reader.
Moderate Points:
1. It would be best to include the logic stated as to why these non-randomized studies were included because authors did not insist on “randomized” studies, just “experimental”.
2. The review may not be complete despite the careful approach to finding stable isotope studies. In my annotation library, I have found some potential journal articles that may be of interest to the authors.
1. 1988 Fomon, Ziegler, et al. Erythrocyte incorporation of ingested 58-iron by infants, Pediatr Res 24, 20
2. Fomon, Ziegler, et al. 1993 Erythrocyte incorporation of ingested 58-Fe by 56-day-old breast-fed and formula-fed infants Pediatr Res 33, 573
3. 1997, Widness, JA, Lombard, KA, Ziegler, EE et al. Erythroctye incorporation and absorption of 58Fe in premature infants treated with erythropoietin, Pediatr Res 1997; 41, 416.
3. To help follow the complex text, it would be best to include more subheadings in section 3.4. and 3.5 and 3.9 to help follow the story.
4. To help follow the discussion, it may be helpful to include a graphical abstract or summary figure to place in the context. Or perhaps key points either in a supplemental “Key Points” from the discussion.
5. The discuss that many of these specific studies in this review are used to generate societies pediatric guidelines. To place in the context of society recommendations, it may be useful to include a table referencing the most recent guidelines for both term and preterm infants for AAP, NASPGHAN, ESPGHAN, referencing these within the table. It may also be useful to then point out the inconsistencies between recommendations and data or need for study.
6. Supplemental files 1, 2 in appendices were not included, so these cannot be address.
Minor Points:
1.Line 40: Preterm infants, born <37 weeks, are especially vulnerable to iron deficiency because they face major challenges in achieving sufficient iron status” rapid growth and lower iron acquisition more appropriate.
2.Line 52: both ferric iron and heme iron are also by receptor mediated endocytosis.
3.There seems like discussing Human milk contains several iron-containing proteins, including lactoferrin, transferrin, ferritin which could be why the little iron contained in milk is absorbed.
Author Response
This review is thoughtful and of much interest to readership. The study logic is clear in its approach to complex topic of interest to societies who generate guidelines. The authors followed traditional PRISMA and other guidelines. There is clinical and scientific need to make the data both complete and understandable for the more casual reader.
- The authors thank you for these comments. We have revised the manuscript to make our review complete and understandable to all readers.
Moderate Points:
- It would be best to include the logic stated as to why these non-randomized studies were included because authors did not insist on “randomized” studies, just “experimental”.
- Thank you for this question. The authors’ goal for this review was to have a complete understanding of iron absorption measured by the gold standard, isotopic method. We did not limit to just “randomized” studies for several reasons. There are scenarios related to iron absorption that ethically cannot be randomized such as breast-feeding vs formulas or the timing of weaning from breastfeeding. We also would like to learn about iron absorption rates at different age intervals. For certain high-risk population such as preterm infants, randomized isotope studies may not be feasible. Even though without randomization, these experimental studies post a risk of selection bias in the causality conclusions, they can still be useful for clinicians and scientists.
- We added an explanation to include non-randomized studies in the first paragraph of the Materials and Methods (lines 76-79).
- The review may not be complete despite the careful approach to finding stable isotope studies. In my annotation library, I have found some potential journal articles that may be of interest to the authors.
- Thank you for bringing this up. We have checked our selection again and found article 1 was included in the original manuscript with a wrong year of 1998. We learned that we excluded the other two articles as “duplicates” due to similar titles. We reviewed all the duplicates and found no other mistakes. We added articles 2 and 3 in the text, tables, and risk of bias assessment.
- 1988 Fomon, Ziegler, et al. Erythrocyte incorporation of ingested 58-iron by infants, Pediatr Res 24, 20
- It is reference #27 in the revised manuscript (lines 166-168).
- Fomon, Ziegler, et al. 1993 Erythrocyte incorporation of ingested 58-Fe by 56-day-old breast-fed and formula-fed infants Pediatr Res 33, 573
- It is reference #29 in the revised manuscript (lines 197-199).
- 1997, Widness, JA, Lombard, KA, Ziegler, EE et al. Erythroctye incorporation and absorption of 58Fe in premature infants treated with erythropoietin, Pediatr Res 1997; 41, 416.
- It is reference #54 in the revised manuscript (lines 154-158).
- To help follow the complex text, it would be best to include more subheadings in section 3.4. and 3.5 and 3.9 to help follow the story.
- We added subheadings for these sections for clarity.
- To help follow the discussion, it may be helpful to include a graphical abstract or summary figure to place in the context. Or perhaps key points either in a supplemental “Key Points” from the discussion.
- Thank you for this suggestion. We added Supplemental Table 3 to summarize the key points. We hope this clarifies the discussion points as well.
- The discuss that many of these specific studies in this review are used to generate societies pediatric guidelines. To place in the context of society recommendations, it may be useful to include a table referencing the most recent guidelines for both term and preterm infants for AAP, NASPGHAN, ESPGHAN, referencing these within the table. It may also be useful to then point out the inconsistencies between recommendations and data or need for study.
- Thank you for this suggestion. We added Figure 2 to illustrate the current recommendations from different societies and also reflect the consistent findings of iron absorption studies.
- Supplemental files 1, 2 in appendices were not included, so these cannot be address.
- We apologize you didn’t see the Supplemental File as a separate file. The revised Supplemental file has Appendix 1 (Research Strategy), Appendix 2 (Risk of Bias Assessment), Supplemental Table 1(Iron Absorption and Red Blood Cell Incorporation), Supplemental Table 2 (Iron Absorption With Different Iron Forms and Complementary Foods), and Supplemental Table 3 (Summary of Iron Absorption Main Findings).
Minor Points:
1.Line 40: Preterm infants, born <37 weeks, are especially vulnerable to iron deficiency because they face major challenges in achieving sufficient iron status” rapid growth and lower iron acquisition more appropriate.
- Thank you for the suggestion. We revised the sentences to reflect this point (lines 40-41).
2.Line 52: both ferric iron and heme iron are also by receptor mediated endocytosis.
- We added in the text to include this information (lines 59-65).
3.There seems like discussing Human milk contains several iron-containing proteins, including lactoferrin, transferrin, ferritin which could be why the little iron contained in milk is absorbed.
- We added to the Discussion to highlight this point (lines 379-381).
Reviewer 2 Report
Comments and Suggestions for Authors
This review article is dealing with many aspects of iron absorption in infants and young children. They selected numerous research papers where authors used isotopes to follow the route of iron in the body. There are chapters in this manuscript about how and what were determined considering iron metabolism, the effect of formulation, milk, additives, forms of the administered iron, EPO, timing and dosage of administration. This means, that the question of iron absorption in children were examined carefully.
The tables are informative and useful. The discussion is not easy to follow. There are a few statements about the certainties of this complex question, and the authors state among limitations that their goal was not to provide guidance to iron administration for children. Still, I miss take-home message, some new information. In the discussion there are still many literatures cited, more diverse facts are shown. I do not get any help at what age of the children what I should do at a certain situation.
Author Response
This review article is dealing with many aspects of iron absorption in infants and young children. They selected numerous research papers where authors used isotopes to follow the route of iron in the body. There are chapters in this manuscript about how and what were determined considering iron metabolism, the effect of formulation, milk, additives, forms of the administered iron, EPO, timing and dosage of administration. This means, that the question of iron absorption in children were examined carefully.
The tables are informative and useful. The discussion is not easy to follow. There are a few statements about the certainties of this complex question, and the authors state among limitations that their goal was not to provide guidance to iron administration for children. Still, I miss take-home message, some new information. In the discussion there are still many literatures cited, more diverse facts are shown. I do not get any help at what age of the children what I should do at a certain situation.
- Thank you for kind feedback and thoughtful suggestions. In response, we revised the Discussion extensively by removing redundant material, reorganizing the points of discussion, and highlighting the similarities and contrasts with other non-isotopic clinical studies. We also added Figure 2 to show the current recommendations on iron supplementation to address your last comment. Figure 2 also highlights consistent evidence on iron absorption.
Reviewer 3 Report
Comments and Suggestions for Authors
The physiology of iron metabolism, especially the absorption of iron in children, is very interesting and an important issue. In their review, Gallahan S et al present a systematic review, which summarizes studies over 60 years to this topic.
This review has some merit because it is a representative synopsis of the literature regarding this issue. The manuscript is well written, however, some formal points should be considered:
· There are too many Tables, which are not really easy to read. Tables should be shortened to a maximum of 4 Tables. Especially Table 2 and Table 6 should be cancelled.
· The section “Discussion” is bloated and should be shortened significantly.
· The Appendix 1 “Search Strategy” of the Supplementary File in its present form is not really helpful for the readership.
Author Response
The physiology of iron metabolism, especially the absorption of iron in children, is very interesting and an important issue. In their review, Gallahan S et al present a systematic review, which summarizes studies over 60 years to this topic.
This review has some merit because it is a representative synopsis of the literature regarding this issue. The manuscript is well written, however, some formal points should be considered:
- There are too many Tables, which are not really easy to read. Tables should be shortened to a maximum of 4 Tables. Especially Table 2 and Table 6 should be cancelled.
- The section “Discussion” is bloated and should be shortened significantly.
- The Appendix 1 “Search Strategy” of the Supplementary File in its present form is not really helpful for the readership.
- Thank you for your appreciation of our review and your suggestions. In response, we moved previously Tables 2 and 6 to the Supplemental file. These tables summarized diverse evidence on iron absorption rates and incorporation and different forms of iron and complementary foods.
- We revised the Discussion extensively to stay more focused on comparing and contrasting the evidence among isotopic (reviewed articles) and non-isotopic clinical studies. As the results, the Discussion is one page less in length and focused on the main findings from isotopic methods from the reviewed articles.
- Appendix 1 “Research Strategy” shows the exact key word search for each database. The readers can copy and paste the entire syntax into the appropriate search box. Each syntax contains the key words that need to be in the titles or abstracts.
Round 2
Reviewer 2 Report
Comments and Suggestions for Authors
The paper has been approved significantly; the changes made are very helpful for understanding the main goal. I accept the answer and the modifications of the manuscript.